# Analysis and Prediction of COVID-19 Multivariate Data Using Deep Ensemble Learning Methods

**DOI:** 10.3390/ijerph20115943

**Published:** 2023-05-24

**Authors:** Shruti Sharma, Yogesh Kumar Gupta, Abhinava K. Mishra

**Affiliations:** 1Department of Computer Science, Banasthali Vidyapith, Tonk 304022, India; gyogesh@banasthali.in; 2School of Technology & Management, SVKM’s Narsee Monji Institute of Management Studies (NMIMS), Indore 452005, India; 3Molecular, Cellular and Developmental Biology Department, University of California Santa Barbara, Santa Barbara, CA 93106, USA

**Keywords:** machine learning, deep learning, data analytics, LSTM, epidemic disease outbreak, COVID-19

## Abstract

The global economy has suffered losses as a result of the COVID-19 epidemic. Accurate and effective predictive models are necessary for the governance and readiness of the healthcare system and its resources and, ultimately, for the prevention of the spread of illness. The primary objective of the project is to build a robust, universal method for predicting COVID-19-positive cases. Collaborators will benefit from this while developing and revising their pandemic response plans. For accurate prediction of the spread of COVID-19, the research recommends an adaptive gradient LSTM model (AGLSTM) using multivariate time series data. RNN, LSTM, LASSO regression, Ada-Boost, Light Gradient Boosting and KNN models are also used in the research, which accurately and reliably predict the course of this unpleasant disease. The proposed technique is evaluated under two different experimental conditions. The former uses case studies from India to validate the methodology, while the latter uses data fusion and transfer-learning techniques to reuse data and models to predict the onset of COVID-19. The model extracts important advanced features that influence the COVID-19 cases using a convolutional neural network and predicts the cases using adaptive LSTM after CNN processes the data. The experiment results show that the output of AGLSTM outperforms with an accuracy of 99.81% and requires only a short time for training and prediction.

## 1. Introduction

Many local governments, like the Indian government, have loosened health-related restrictions as the global coronavirus pandemic approaches its 3rd year. The majority of people are currently thought to be immune to SARS-CoV-2 due to vaccination or spontaneous infection, but data and experience show that new variants are evolving, leading to limited outbreaks and unexpected consequences for preventative and therapeutic methods. Humanity has contracted a virus at a time when people are breaking new ground in technology and battling the problem of the climate catastrophe. The coronavirus infection epidemic is now categorized as a pandemic by the World Health Organization (WHO), and new variants with varied degrees of severity may manifest themselves every season. A virus’s ability to spread depends on its susceptible sources, viral latency, and susceptibility [1]. Human life and civilization could be seriously threatened by the development of this disease [2,3]. When a disease outbreak starts, the ways it is evaluated are very important for taking quick steps to stop the disease from spreading. Each epidemic in a country or province usually grows at different rates over time. This is because of things like seasonal changes and how the virus changes over time [4].

Over 600 million instances of COVID-19 infection brought on by new strains have been reported globally since the pandemic started. Unsettlingly, things could perhaps get worse in the future. A recent study found that the virus-infected survivors had a high rate of neuropsychiatric issues. Similarly, even 2 years after diagnosis, individuals with COVID-19 are more likely than those with other respiratory illnesses to develop neurological and psychiatric disorders such as dementia, psychosis, etc. [5]. For their analysis, another study examined the data of around 1.28 million people who received a COVID-19 diagnosis between 20 January 2020 and 13 April 2022. The researchers contrasted their findings with those of a similar group of people who had other respiratory conditions. The study looked at 14 mental and neurological issues. Adults were shown to have a higher risk of mental illnesses or anxiety after COVID-19; however, this risk returned to baseline levels within two months for people with other respiratory infections. Even 2 years after the original infection, the risk of cognitive impairment, often known as brain fog, dementia, psychotic diseases, and epilepsy or seizures, remained high. Minors, adults between the ages of 18 and 64 years, and seniors aged more than 65 years were the age groups into which the studied patients were divided by the researchers [6].

### 1.1. Increased Prevalence of Dementia in COVID-19-Infected Individuals

Those over 64 with COVID-19 infection exhibited a greater frequency of cognitive abnormalities (15.4%) among the principal problems than those with other respiratory illnesses (12.3%). Similarly, the COVID-19 infection raised the risk of dementia by 1.2 percentage points in the same age range. The study finds that elderly people who are infected are more likely to experience insomnia, mental difficulties, etc. [7].

### 1.2. Threat by Different Variants

The neurological and mental outcomes during the delta and omicron waves were equivalent, according to the study, which also assessed the possibility of illnesses throughout the development of several COVID-19 variations [7]. Also, 2 years following the COVID-19 infection, it is more likely that issues like dementia, cognitive impairment, etc., will be identified. According to a previous study by the same group at Oxford University, mood disorders, strokes, or dementia were present in COVID-19-infected people six months after infection. A further study that was published in the Lancet journal last year found that COVID-19 was responsible for a spike in major depressive disorder and anxiety disorder worldwide [8].

### 1.3. Technological Advancements

People have been given technology, and if we use it wisely, we can at least help doctors and government officials fight conditions that are like pandemics. One such technological innovation that could be useful during the pandemic is the forecasting and prediction of the infection condition beforehand. In the past ten years, machine learning (ML) has become an important area of study because it has been used to solve numerous types of real-world issues that are very difficult and complicated [9]. The basic flow of machine learning approach is depicted in Figure 1. The World Health Organization (WHO) officially refers to the novel coronavirus SARS-CoV-2, also known as COVID-19, and researchers created an early warning model for its transmission [10]. Over the world, COVID-19 has presented a serious threat and affected many human lives. The virus, which has been around for two years, is still having an impact on our lives and comes in a variety of forms with many symptoms that are particularly challenging to list all at once. The management, governance and readiness of the healthcare department and its suppliers, and ultimately the prevention of disease spread, depend on accurate and effective models [11]. The major goal of this research in this regard is to develop a model that can forecast COVID-19-positive cases. All shareholders will benefit from this while developing and revising their pandemic response plans. For precise COVID-19 spread prediction, the research recommends an adaptive gradient LSTM (AGLSTM) model utilizing multivariate time series data. The RNN (Recurrent Neural Network), LSTM (Long Short Term Memory), LASSO (Least Absolute Shrinkage and Selection Operator), Ada-Boost (Adaptive Boosting), Light Gradient Boosting, and KNN (K- Nearest Neighbors) models are also used in the research to successfully and reliably predict the spread of this terrible disease.

This research aims to fulfill the following objectives:Research and analysis of COVID-19 multivariate data;To come up with a general method for predicting the COVID-19 outbreak that is based on data and machine learning;To compare and evaluate how well different prediction methods, such as LASSO, Ada-Boost, Light Gradient Boosting, KNN, RNN, LSTM, and Adaptive Gradient LSTM, can predict the number of deaths, the number of positive cases, and the number of recovered cases;To benefit from a potent activation function that helps obtain the best performance;To profit from the established advantages of deep learning techniques in processes that assist in epidemic disease outbreaks and health decision-making.

The research suggests using an adaptive gradient LSTM (AGLSTM) model with multivariate time series data to accurately predict how COVID-19 will spread. The framework of the model is presented in Figure 2. In the study, models including LASSO, Ada-Boost, Light Gradient Boosting, KNN, RNN, and LSTM are also used to forecast the spread of this contagious disease. The proposed models were carefully analyzed using a sizable multivariate COVID-19 dataset and the entire workflow of the proposed model is depicted in Figure 3. Our experimental results show the higher performance of the proposed models.

Due to how complicated the COVID-19 outbreak is, its uncertainty, and how many countries do not have important data because they do not have as many ways to collect data as a country like India, the main concern is not only how accurate the models are, but also how well they can be used in a wide range of situations [12]. This work attempts to solve this complex problem in this environment with the least amount of training and prediction time possible. By reusing previously developed prediction models, we also take data fusion and transfer learning into consideration. This is because; deep neural networks and deep ensemble learning training require large amounts of computational time, massive data, and computer resources. In fact, the focus of this work is data fusion, a method that integrates data to create knowledge that is more accurate, consistent, and informative, while real data could be inaccurate, unclear, inconsistent, and insufficient [13]. We can develop precise predictions that might run into problems when gathering COVID-19 data by concentrating on data fusion.

The article is organized in the following manner. Section 2 discusses the research works that employ machine learning and deep learning models for the prediction of COVID-19. The proposed state-of-the-art using multivariate data for the prediction of COVID-19 is outlined in Section 3, which will be followed by the experimental setting used in this research. Subsequently, the results of the conducted trials are given and discussed, and the article ends with a conclusion, future work and limitation section.

## 2. Literature Review

This section’s main goal is to assess some of the most significant recent relevant attempts at COVID-19 outbreak prediction utilizing machine and deep learning methods. This review of the literature focuses, in particular, on studies that forecast everyday events that are verified or positive. M. Li et al. [14] suggest a machine learning technique for estimating the daily numbers of cumulative confirmed cases, newly confirmed cases, and death cases of COVID-19 in China from 20 January 2020 to 1 March 2020, using data from the National Health Committee of China. A comparison of machine learning and soft computing models for COVID-19 outbreak prediction in five counties was conducted by S.F. Ardabili et al. [15]. Two machine learning models’ results—the multi-layered perceptron and the adaptive network-based fuzzy inference system—were promising and had a high capacity for long-term prediction. S. Bandyopadhyay et al. [16] used recurrent neural networks (RNNs) to predict COVID-19 confirmed (positive), negative, released, and death cases. RNNs can represent the prediction of temporal (sequential) data. Three models—a combined LSTM-GRU model, a gated-recurrent unit (GRU) model, and a long short-term memory (LSTM) model—were presented. According to experimental findings on the COVID-19 dataset for South Korea from 20 January 2020 to 12 March 2020, the combined model achieves the highest level of accuracy. A convolutional neural network (CNN) model was proposed by C.J. Huang et al. [17] to predict the number of COVID-19 verified cases in China from 23 January 2020 to 2 March 2020, using information from Growing News Network and WHO. According to experiments, the recommended CNN model outperforms MLP (multilayer perceptron), LSTM, and GRU. For predicting the number of novel coronavirus (COVID-19)-positive reported cases for 32 Indian states and union territories, P.H. Kumar et al. [18] used deep learning-based models, specifically LSTM variants such as deep LSTM, convolutional LSTM, and bi-directional LSTM models. They found that bi-directional LSTM gave the best results, while convolutional LSTM gave the worst. For predicting the number of new and recovered cases for six countries—Italy, Spain, France, China, the United States, and Australia—A. Zeroual et al. [19] presented a comparison of five deep learning models (basic RNN, LSTM, Bidirectional-LSTM, gated recurrent units (GRUs), and VariationalAutoEncoder (VAE)). Their results demonstrated the VAE’s superior performance over the other methods and the deep learning models’ promising potential in forecasting COVID-19 cases. A. Hernandez-Matamoros, et al. [20] made a way to run and analyze the ARIMA model for 145 countries spread out over 6 continents. The goal was to link countries in the same area so that the spread of the virus could be predicted. S. Chae et al. [21] previously compared DNN and LSTM models to the auto-regressive integrated moving average (ARIMA) for the prediction of infectious diseases, and the findings showed that DNN and LSTM models outperformed ARIMA. Table 1 summarizes a quick comparison study and review of various deep learning models. This validates our choice to use deep learning techniques, which have been shown to be precise and efficient in predicting COVID-19 outbreaks. The study’s main contribution is the creation of a standardized, data-driven, accurate, and generic COVID-19 outbreak prediction technique. In the part after this one, we will discuss the models’ historical contexts. It is discovered that machine learning is a useful method for simulating the COVID-19 epidemic due to its highly complex structure and variance in behavior from country to country.

## 3. Materials and Methods

### 3.1. Data Set Description and Data Preparation

In this study, real-time observations are incorporated for up-to-date analysis and for the prediction of COVID-19 results. There are 2 types of datasets used in this study: (1) The global dataset from January 2020 to August 2021, which is being gathered from covid19india.org and is available via the online source Kaggle. Three separate time series datasets were gathered, including confirmed, recovered, and death cases. It also includes information such as the name of the province, country, and the number of cases by date. (2) Second, data for COVID-19 is collected from Indiastathealth.org, which includes parameters like confirmed cases by date, confirmed deaths, vaccination, policy responses, mobility, generic, hospitalizations, discharged or migrated, and the number of Asha deaths.

The panda profiling feature is used for Exploratory Data Analysis, and the interaction between 3 important features is shown in Figure 4. Correlation factors like Spearman’s (ρ), Pearson’s (r), and Kendall’s (τ) are used to do the statistical analysis, which is presented in Figure 4a–c. Pearson’s correlation specifies the linear correlation while testing the similarities in the ordering of the data; when it is ranked by quantity Kendall’s correlation is used, which is highly reflected in the below variables. Also, Spearman’s correlation specifies the strength and direction of the association between ranked variables. From the above correlation factors, we can see that the variables are highly correlated linearly and in terms of quantity, they have similarities. The scatter diagrams for confirmed, active, and cured cases and deaths are depicted in Figure 5.

Real-world datasets can be unreliable, and studying raw data might lead to incorrect conclusions. As a result, data must be pre-processed before being analyzed. There are a variety of pre-processing approaches available to deal with messy data in order to ensure consistency in knowledge discovery data [34]. Multiple files’ properties can be concatenated to make a single file in a usable format [35]. Data reduction procedures can be used to reduce the number of attributes by reducing redundancy in the dataset [36,37].

The support and resistance levels are determined by technical analysis indicators. The support level indicates when the number of cases has decreased, and the resistance level indicates when it has risen [38]. They aid in recognizing both upward and downward trends. To extract noise-free features from the existing raw features, the TA-lib software is used to reveal significant patterns. The indications utilized to execute feature engineering are listed here [39]:

SMA (simple moving average): The average of a chosen range of cases is determined by the number of periods in that range.

Weighted moving average (WMA): The formula for calculating the weighted moving average (WMA) is to multiply the current cases by the corresponding weights and then add the results.

The exponential moving average (EMA): It is a sort of weighted moving average that emphasizes current case data, but the rate of decline between one case and its prior case is not linear but rather exponential.
(1)$$EMA=Cases\left(t\right)*k+EMA\left(y\right)*\left(1-k\right),$$
where t is today, y is yesterday, N is the number of days in EMA (i.e., the smoothening range), and k = 2/(N + 1)

After feature engineering, certain undesirable features were deleted using linear interpolation. The feature selection method is depicted in Figure 6 where missing value imputation was applied when necessary after dividing all features into blocks for each set of periods (smoothing range) and technical indicators. Then, using a random forest regressor as an estimator, each set is fitted with Recursive Feature Elimination, Cross-Validated (RFECV). The most significant characteristic is chosen by RFECV after the features are ranked. At each iteration, the step size is reduced to 0.6, removing 60% of the least significant features. From each block, the feature with the highest rank is selected. The Variance Inflation Factor (VIF) is used to leave out characteristics that are strongly linked to other independent characteristics.

After preparing the dataset, all date columns are converted to data-time objects to group the data by ‘Date’ to find the cumulative sum of cases. The description and time series plotting of 3 variables, cases, deaths, and cured, are shown in Table 2 and Figure 7, respectively. Then re-sampling the number of cases is done on a monthly and weekly basis, and is shown in Figure 8 and Figure 9. Also, the time series visualization for 200 days is shown in Figure 10. Then we set up helper functions for forecasting, extracting the last n days from the time series and plotting the last n days from the time series. Afterward, multivariate data is prepared with a Keras format series, which is used to convert the numpy series into a 3D form, and then data splitting is done for training and testing.

### 3.2. Machine Learning Models

#### 3.2.1. Lasso Regression

Lasso (Least Absolute Shrinkage and Selection Operator) Regression is a sort of regularized linear regression with an L1 penalty that is widely used. This causes the coefficients for input variables that don’t contribute much to the prediction task to diminish. This penalty allows some coefficient values to be set to 0, thereby removing input variables from the model and allowing for automatic feature selection. Lasso Regression is a linear regression extension that includes a regularization penalty in the loss function during training [40].

#### 3.2.2. K-Nearest Neighbor

The k-NN method is one of the most basic classical machine learning algorithms. Its first application was in classification. For unlabeled samples, the k-NN technique finds the k-closest examples among all the labeled cases and predicts the class of the unlabeled ones based on their majority class [41]. The examples are described by a vector of features, and their similarity is given by a distance function, commonly the Euclidean distance. The k closest cases to the unlabeled case are thus the k nearest neighbors utilized to categorize it according to the vector of features and the distance function. The k-NN may easily be used to perform regression. The target variable is numerical in this example. When the target variable is unknown, the k-NN method attempts to locate the k-closest neighbors among the set of inputs whose target value is known. Either the mean or the median is the expected goal value.

#### 3.2.3. Ada Boost

Ada Boost (Adaptive Boosting) is an ensemble learning method (sometimes known as “meta-learning”) that was originally devised to improve the effectiveness of binary classifiers. Ada Boost employs an iterative strategy to improve poor classifiers by learning from their mistakes. Ada Boost is a prominent boosting technique that seeks to construct a strong classifier by merging many weak classifiers [42]. A single classifier may not be able to reliably forecast an object’s class, but we can develop a powerful model by combining numerous weak classifiers, each learning from the others’ incorrectly categorized objects. A weak classifier is one that outperforms random guessing but still has trouble assigning classes to objects [43].

#### 3.2.4. Light Gradient Boosting Machine

This is also a type of gradient boosting, with light denoting a lighter form. This is thought to make the model more efficient, faster, and more accurate. LGBM stands for light gradient boosting machine and is a type of gradient boosting. Light GBM, like other gradient-boosting techniques, is based on Decision tree methods. We can reduce memory utilization and boost efficiency with the help of Light GBM. The primary distinction between Light GBM and other gradient boosting frameworks is that Light GBM grows leaf-wise rather than horizontally. The other algorithms, on the other hand, extend horizontally in a level-by-level manner. The leaf with the least error and highest efficiency is chosen by Light GBM. This strategy is far more effective at lowering the error rate [44]. In other words, it expands leaf-by-leaf, whereas others expand level-by-level, and its architecture is shown in Figure 11.

#### 3.2.5. Recurrent Neural Network

RNNs are deep learning models that are often used to tackle problems involving sequential input data, such as time series. RNNs are a sort of neural network that remembers what it has processed previously and can thus learn from past iterations during training. “A recurrent neural network (RNN) is a type of artificial neural network in which nodes are connected in a directed graph that follows a temporal sequence. This enables it to behave in a temporally dynamic manner. RNNs, which are derived from feed-forward neural networks, can process variable-length sequences of inputs by using their internal state (memory). Because its connections create a directed cycle, a Recurrent Neural Network (RNN) deals with sequence problems. In other words, they can keep the state from one iteration to the next by feeding the next step their own output. Only short-term memory can benefit from a simple recurrent neural network. If we have a longer time dependency, we will find that it has a basic flaw (vanishing/exploding gradient) [45,46].

#### 3.2.6. Long Short-Term Memory (LSTM)

Long short-term memory is a gated memory unit for neural networks. Due to its ability to learn additional parameters, the LSTM cell increases long-term memory in a way that is even more efficient. As a result, it is the most effective [Recurrent Neural Network] for predicting, particularly when your data show a longer-term trend. LSTMs are state-of-the-art models for forecasting at the moment. The memory’s contents are managed by three gates. These gates are basic logistic functions of weighted sums that can be learned using back-propagation. It means that, despite its complexity, the LSTM fits into the neural network and its training process perfectly. It is capable of learning what it needs to learn, remembering what it needs to know, and recalling what it needs to recall without any additional training or optimization. The cell state (4), or long-term memory, is managed by the input and forget gates (1) and (2), respectively. The output gate (3) generates the concealed state (5), which is the memory targeted for usage. This memory structure allows the network to remember for a long period, which is a feature that was previously lacking in traditional recurrent neural networks [47].
(2)$$i_{t}=sigmoid(W_{i}x_{t}+U_{i}h_{t-1}+b_{i})$$
(3)$$f_{t}=sigmoid(W_{f}x_{t}+U_{f}h_{t-1}+b_{f})$$
(4)$$o_{t}=sigmoid(W_{o}x_{t}+U_{o}h_{t-1}+b_{o})$$
(5)$$c_{t}=f_{t}c_{t-1}+i_{t}tanh(W_{c}x_{t}+U_{c}h_{t-1}+b_{c})$$
(6)$$h_{t}=o_{t}tanh(c_{t})$$
where $i_{t}$ represents the input gate, f_t_ represents the forget gate, o_t_ represents the output gate, c_t_ represents the cell state (memory) at timestamp t, and h_t_ represents the hidden state that is the output of the previous LSTM block. W_i_, W_f_, W_o_ and U_i_, U_f_, and U_o_ refer respectively to the weight parameters, and b_i_, b_f_, and b_o_ denote the bias parameters. W_c_, U_c_ denotes weight parameters, b_c_ is the bias parameter, and o refers to the element-wise multiplication.

#### 3.2.7. Adaptive Gradient LSTM

In this instance, 500 data point-sized sliding windows are employed. The initial 400 points are reserved for training, while the remaining points are used for model evaluation. Scaling features is among the most important preprocessing steps. The standard scalar uses the mean to scale the data, and as the mean is susceptible to outliers, the presence of outliers will influence the scaling. After removing outliers with the Robust Scale, we employed a min-max scale. The robust Scale is unaffected by a small number of extremely large marginal outliers because it is based on percentiles. Either a robust scale followed by a min-max scale or a performance-based standard scale is utilized here.

##### Model Structure

The proposed model network structure is designed as AGLSTM, and the role of each module is to capture the complex situation of COVID-19 cases. Initially, our algorithm takes the samples of a batch of tasks from all the training/prediction tasks, and then the extracted tasks are divided into training and testing data. The goal of the training dataset is to calculate the optimal parameters for each task, whereas the test dataset is to calculate the optimal parameters for the whole model. Both datasets pass through the same network structure. The input training data will be processed by the CNN first. The CNN is used to capture the numerous influences of all variables, and it is also capable of integrating the spatial relationship between the data, making feature extraction more convenient. The entire sequence is passed to the LSTM layer. Since the observation data is primarily time series data, the CNN’s output feature maps are input into the LSTM to learn the sequence’s long-term dependencies. After that, the input of LSTM is transferred to the fully connected layer, which outputs the prediction result; gradient descent is used to minimize loss and compute its value. Finally, the optimal assignment parameters are determined. For the input data of the test data, the input network structure remains unaltered, while the network’s parameters are optimized for the task. On this basis, gradient descent is repeated in an effort to minimize loss and increase accuracy. Finally, we obtain the model’s relative optimal parameters. The proposed methodology and the network structure are shown in Figure 12.

### 3.3. Performance Metrics

We examined the outcomes of the aforementioned trials using a variety of metrics, including precision, mean squared error (MSE), mean absolute error (MAE), mean absolute percentage error (MAPE), R-squared (R^2^), and root mean squared error (RMSE) [33]. Accuracy facilitates the calculation of how frequently the forecast matches the actual label. MAE and MSE are utilized to calculate the mean absolute error value and mean squared error between y true and y predicted, respectively. MAPE is the measure of the prediction accuracy of a statistical technique, such as trend estimation, used for forecasting. The calculation is as follows:(7)$$\mathrm{MAE}=\frac{1}{N}\sum_{n=1}^{N}|{Ỹ}_{n}-Y_{n}|$$
(8)$$\mathrm{RMSE}=\sqrt{\frac{1}{N}\sum_{n=1}^{N}{\left({Ỹ}_{n}-Y_{n}\right)}^{2}}$$
(9)$$\mathrm{MSE}=\frac{\sum_{n=1}^{N}({{Ỹ}_{n-}Y_{n})}^{2}}{N}$$
(10)$$\mathrm{MAPE}=\frac{\sum_{n=1}^{N}\left|\frac{{Ỹ}_{n-}Y_{n}}{{Ỹ}_{n}}\right|}{N}*100$$
where ${Ỹ}_{n}$ represents the number of COVID-19 predicted cases by the model and $Y_{n}$ represents the observed value of the actual COVID-19 cases. N is the data size that needs to be predicted.

The Deep learning models are assessed using various performance measures like accuracy, precision, recall, F1-score, and support. Overall accuracy is calculated as the total true findings divided by the total number of samples. Sensitivity and specificity are concepts used to characterize the true positive and true negative rates. The formulas for computing these performance characteristics are offered by Equations (11)–(14).
(11)$$Accuracy=\frac{TP+TN}{TP+TN+FP+FN}$$
(12)$$Recall=\frac{TP}{TP+FN}$$
(13)$$Precision=\frac{TP}{TP+FP}$$
(14)$$F1\;Score=2\times \frac{Precision\times recall}{Precision+recall}$$

### 3.4. Experimental Setup

In this study, after creating the Ada-Boost, KNN, LGBM and Lasso models, they are tuned with hyper-parameters with a learning rate of 0.05, the loss is linear, estimators are set at 90, and probability threshold values are verbose as true and with random states. These hyper-parameters tuned models are then finalized with 10 cross-fold validations. For hyper-parameter tuning, a random grid search of hyper-parameters over a predefined search area is utilized. R2 is modified using the optimized parameter in order to optimize it. When determining the optimal production model, metrics are not the only criterion to consider. Other parameters, such as training time and the standard deviation of k-folds, are also evaluated.

Furthermore, we looked at the best parameter choices for RNN, LSTM and our proposed model AGLSTM, such as the number of epochs, batch size, and neurons, to get a decent prediction result for COVID-19 cases and death. The following are the descriptions of these parameters:Epochs: the number of epochs is a parameter that specifies how many times the learning technique will run over the entire training dataset. The number of epochs refers to the number of full passes over the training dataset;Size of the batch: the batch size is a parameter that specifies the number of samples to work with before updating the internal model’s variables. The batch size relates to how many samples are processed before the model is updated;Number of neurons: the number of neurons in a network affects its learning capacity. In general, the more neurons there are, the faster the issue structure is learned at the expense of a longer learning period. With increased learning capability comes the risk of over-fitting the data utilized for training [32].

The results reveal that for RNN and LSTM, 40 epochs are sufficient, whereas, for AGLSTM, 150 epochs are sufficient. This indicates that the training process has stabilized and that increasing the number of epochs is no longer beneficial. We calculated and tracked the validation and training losses. If the validation loss rises, over-fitting is a possibility. To avoid over-fitting, we should increase the number of epochs as much as possible. The pseudo-code for the proposed algorithm is mentioned in Algorithms 1 and 2.
**Algorithm 1:** Base Algorithm **Input:** Load dataset for pre-processing **Output:** Positive COVID-19 cases and deaths over n days**Normalize** the dataset into values from 0 to 1**Initialize** the sequential network**Set** the no. of RNN blocks and input the activation function**Select** the training window size**for** n epochs and batch size, do**Train** the network**end** for**Run** predictions**Calculate** the loss function, accuracy

**Algorithm 2:** Ada_Gradient_LSTM**Input:** Initialize sequential model with dataset passed through CNN **Output:** Positive COVID-19 cases and deaths over n days**Adding** the first LSTM layer and some Dropout regularization**Set** LSTM units = 45, return_sequences = True, input_shape = (X_train.shape [1],1)**Select** Dropouts as 0.2**Adding** the second LSTM layer and some Dropout regularization**Set** LSTM units = 65, return_sequences = True**Select** Dropouts as 0.2**Adding** the third LSTM layer and some Dropout regularization**Set** LSTM units = 85, return_sequences = True**Select** Dropouts as 0.2**Adding** the fourth LSTM layer and some Dropout regularization**Set** LSTM units = 128**Select** Dropouts as 0.2**Adding** the output layer**Set** Dense units = 1**Calculate** the loss function/optimization strategy and fit**Select** optimizer as adam**Fit** the desired number of passes over the data (epochs)**Set** train and test epochs, Batch_size = 64 and verbose = 1**Return** results

## 4. Results

On the multivariate data set for COVID-19 in India, we used LASSO Regression, KNN, Ada-Boost, Light gradient boosting, RNN, LSTM, and our suggested algorithm Ada-GLSTM. Despite the fact that all of the methods discussed above performed well, our proposed approach outperformed them all. First of all, Lasso, KNN, Ada-Boost and light gradient boosting are implemented with 10 cross-folds and extra trees regressor estimator and attempted the results of the various metrics used for determining the efficacy of the various prediction models as shown in Table 3. These created models are then tuned with hyper-parameters, fitting 10 folds for each of the 10 candidates, totaling 100 fits, and then the results are shown in Table 4 and Table 5. The residual plots and prediction error plots for Ada-Boost, KNN, LGBM and LASSO are depicted in Figure 13a–d and Figure 14a–d, respectively.

The simple RNN model was initially constructed; however, it was quickly abandoned because of its low performance. The COVID-19 cases produced from the datasets were then predicted using the LSTM-based prediction model. The LSTMs are made up of cell states that actively forget or remember data. The forget gate, input, and output gates were the three gates that worked in the cell state. As a result, we used these gates to create three layers for the LSTM model: the LSTM layer, the Dropout layer, and the dense layer. In comparison to a simple LSTM model, we implemented two different phases here. For reproducibility, we started with a fixed random seed and a rectified linear activation function (ReLU). We approximated the Keras metrics for the AGLSTM using the ReLU and found the best results. In addition, we reduced the vanishing gradient point inaccuracy. Additionally, the models RNN, LSTM and AGLSTM with results are shown in Table 6. The loss and accuracy diagrams for RNN, LSTM and AGLSTM are shown in Figure 15a–f, respectively.

## 5. Discussion

According to the data obtained in the previous section, it became apparent that the technique utilized in the proposed prediction model (AGLSTM) produced an accuracy of 99.81 percent. This result is superior to the other models; although the other models also gave good accuracy, our proposed model outperformed not only in terms of accuracy but also took minimum time to execute also with minimum network bandwidth. Moreover, we have implemented LASSO, Light gradient boosting algorithm, KNN, and Ada-Boost and scored the R^2^ value as 0.9215, 0.9354, 0.9321, and 0.8999, respectively. RNN and LSTM resulted in sufficient accuracy for predicting the number of cases as 96.95 and 97.97 percent. In addition, we discovered that the proposed AGLSTM model yielded superior results with the presence of the CNN module before being added to the LSTM model, and then the optimal parameters were identified through the fully connected layer. Also, due to the presence of three types of memory in LSTM, the first being the Input Gate, which determines which values from the input are used to update the memory state (take the input from tanh and input weight and apply the RELU activation, then the output 0 or 1). Second, the Forget Gate determines which data is discarded from the block. The third component is the Output Gate, which determines the output based on the input and the block’s memory. In order to improve the forecast results and what was concluded during the analysis, we discovered through a literature review that the process of removing noise from any data depends on time (time series), as many researchers did not pay particular attention to this and ignored this step; consequently, some inaccurate results may be produced. This piqued our interest and inspired us to propose a model which eliminate the confusion in the data, resulting in more accurate experimental results.

## 6. Conclusions

As the global coronavirus pandemic enters its 3rd year, some local governments have lifted prohibitions on public health. Currently, it is thought that the majority of the population is immune to SARS-CoV-2 through vaccination or spontaneous infection, but experience and statistics show that new variants will develop, causing local outbreaks and having unforeseeable effects on preventative and treatment measures. There is currently no cure for this disease, and the likelihood of accurately predicting its severity is minimal. In order to make predictions concerning this disease, machine learning models have been implemented. For this objective, we proposed a DL-based prediction model using time series datasets for India. This study proposed an AGLSTM model based on a framework where a network structure combines CNN and gradient LSTM with fully connected layers for the prediction of COVID-19 deaths and cases. The model consists of two parts; firstly, CNN captures the features of the input data and combines them to form high-level data features, which are then fed to the LSTM model. The resultant from the LSTM model then passes through the fully connected layer, which helps in reducing the loss and adaptive gradients are calculated for optimizing the parameters. Experimental results show that our proposed AGLSTM model outperforms other models in terms of both accuracy and TT (s) to execute the model. In terms of interpretability, multivariate influencing factors, and gradient updates, the AGLSTM model still has space for improvement. For comparison, the KNN, LASSO, LGBM, Ada-Boost, RNN and LSTM prediction models were assessed. We used the Python programming language to create and construct models. The suggested AGLSTM prediction model accurately predicted the number of confirmed COVID-19 cases and deaths with an accuracy of 99.81%. This model also reduced the error value at the point of vanishing gradient. Future plans include expanding this model to forecast the amount of COVID-related cases and deaths in each country.

## 7. Limitations

In order to forecast the growth rate of COVID-19 cases and the rate at which patients recover from the virus in different states, Machine Learning techniques are used to make these forecasts. The study is limited to an analysis of the influence of COVID-19 on the Indian dataset, and the model can be assessed against the datasets of other nations in order to estimate the cured and death rates. The performance of the model can be assessed using alternative feature sets. From the experimental data, it can be stated that the proposed AGLSTM model attained the maximum accuracy, followed by RNN, LSTM and fine-tuned ensemble models. Machine Learning technologies aid in predicting the ongoing development of the COVID-19 pandemic by extracting information regarding the virus’s epidemiological pattern. Future work can be expanded for patient-specific tailored healthcare leveraging the Internet of Things and Machine Learning. In the future, models can be created to forecast respiratory illness infection patterns, virus variations, and peak levels in addition to cumulative reporting such as confirmed, new, and fatal COVID-19 cases. With the combined efforts of society, science, and technology, the COVID-19 outbreak is manageable if comprehensive and stringent disciplinary control measures are implemented. The scope of the current investigation is confined to the contribution of epidemiologic expertise to the evaluation of the analytical model’s performance; the same may be considered when evaluating future studies.

## Figures and Tables

**Figure 1 ijerph-20-05943-f001:**
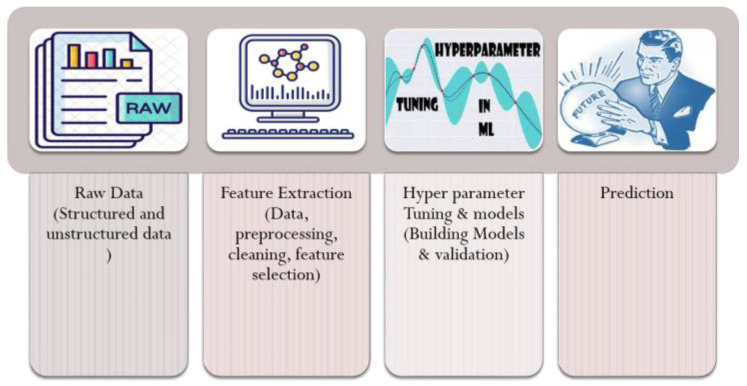
Machine Learning Approach.

**Figure 2 ijerph-20-05943-f002:**
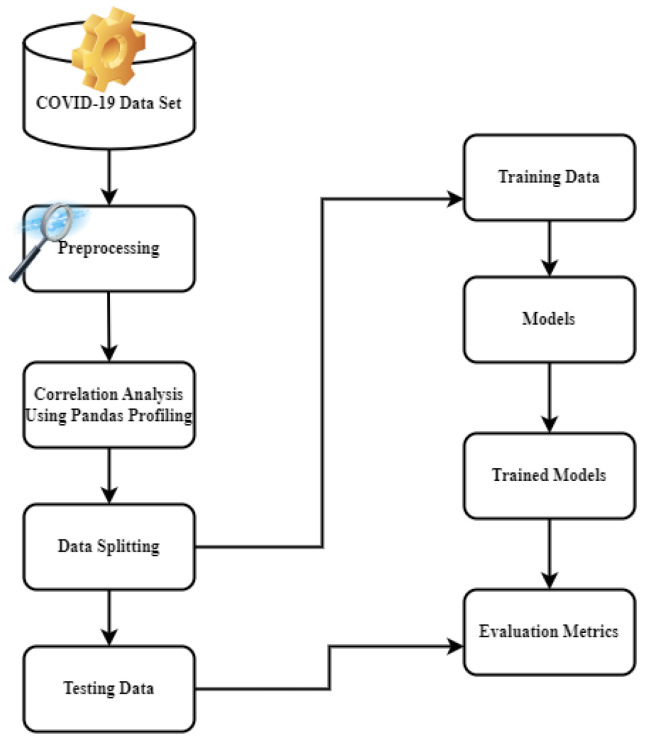
Framework of the model.

**Figure 3 ijerph-20-05943-f003:**
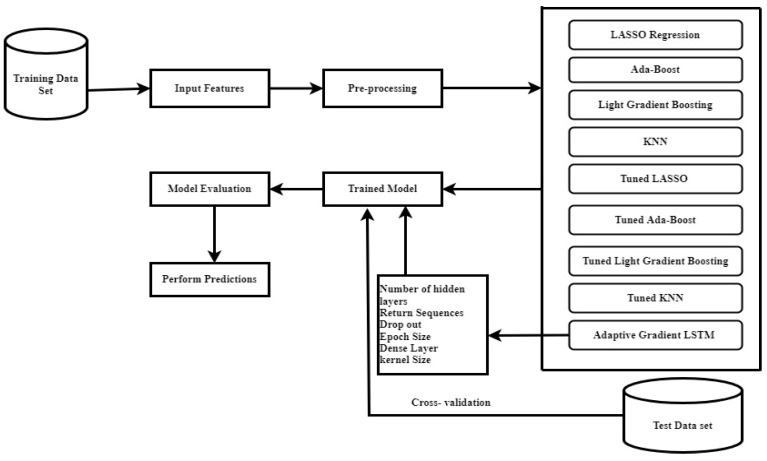
Workflow of ensemble and proposed models.

**Figure 4 ijerph-20-05943-f004:**
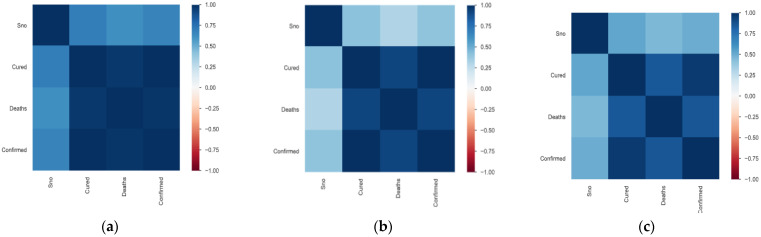
(**a**) Spearman’s, (**b**) Pearson’s, and (**c**) Kendall’s correlations.

**Figure 5 ijerph-20-05943-f005:**
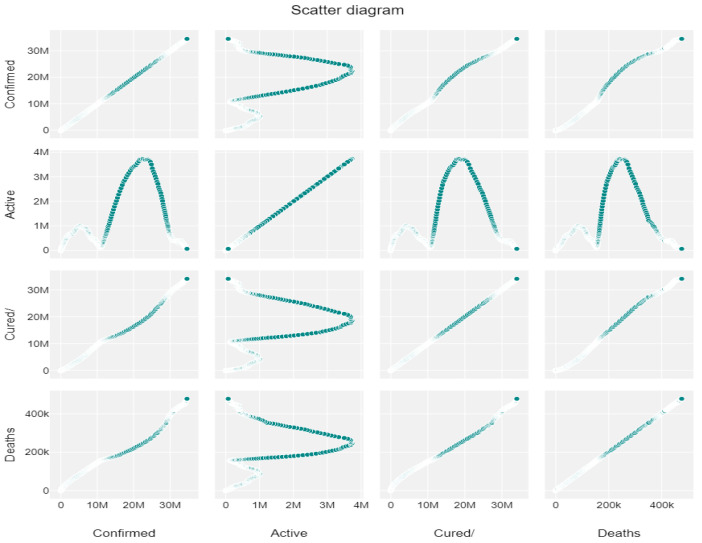
Scatter diagrams of confirmed, active, and cured cases and deaths.

**Figure 6 ijerph-20-05943-f006:**
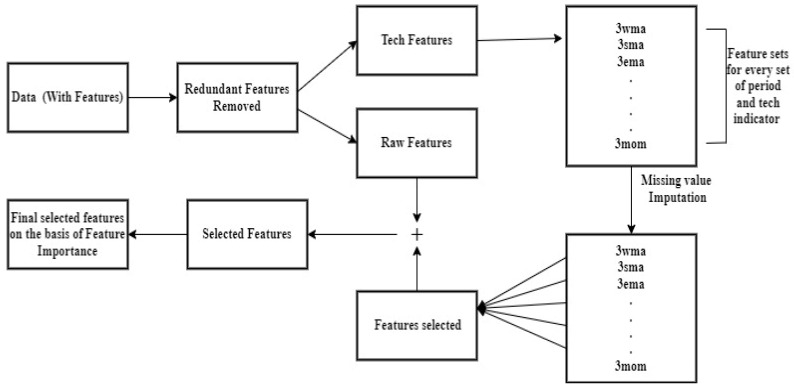
Feature Selection Process.

**Figure 7 ijerph-20-05943-f007:**
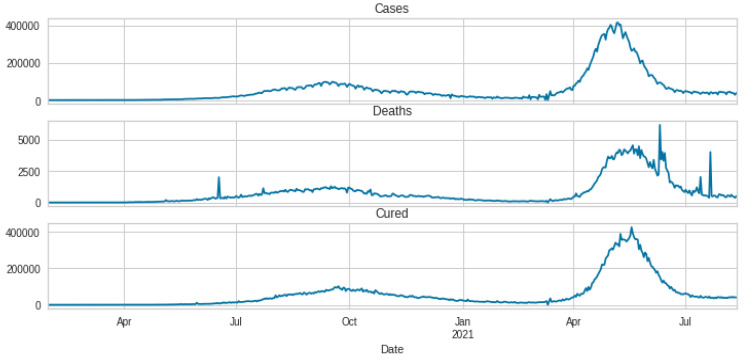
Time series plotting of three individual features.

**Figure 8 ijerph-20-05943-f008:**
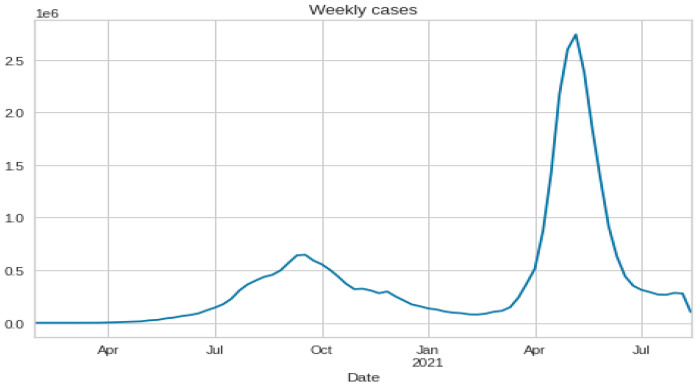
Re-sampling the number of cases on a weekly basis.

**Figure 9 ijerph-20-05943-f009:**
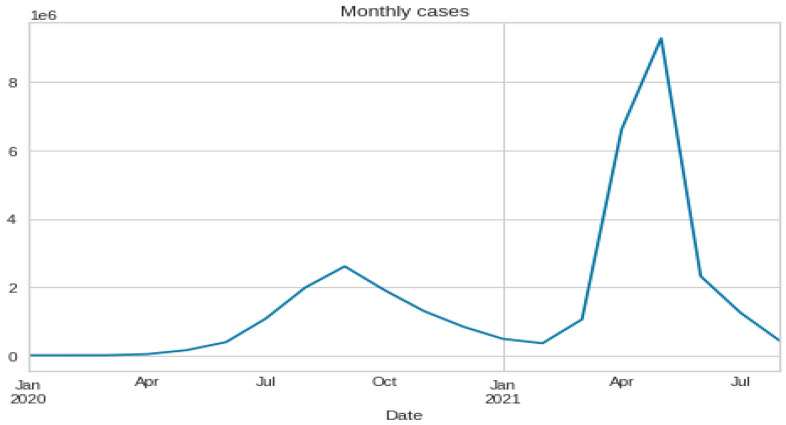
Re-sampling the number of cases on a monthly basis.

**Figure 10 ijerph-20-05943-f010:**
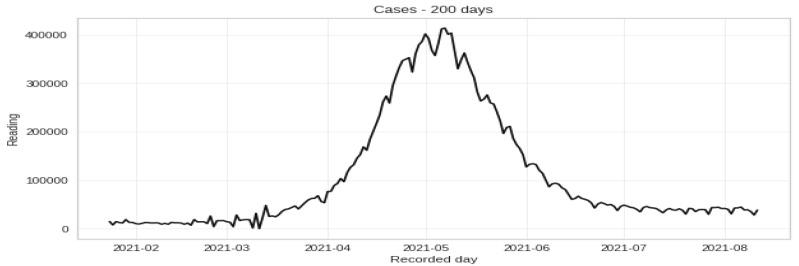
Time series data visualization for 200 days.

**Figure 11 ijerph-20-05943-f011:**
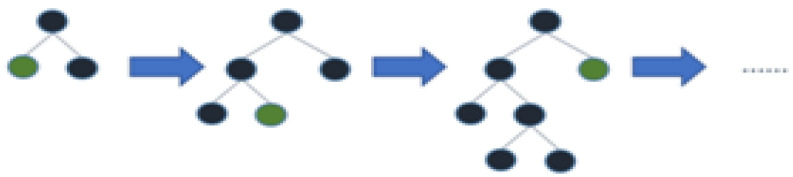
Leaf-wise tree growth architecture of Light gradient boosting machine algorithm.

**Figure 12 ijerph-20-05943-f012:**
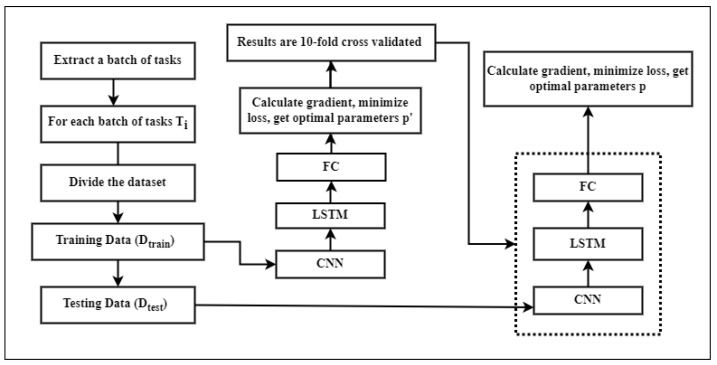
Methodology and internal structure of our proposed model AGLSTM.

**Figure 13 ijerph-20-05943-f013:**
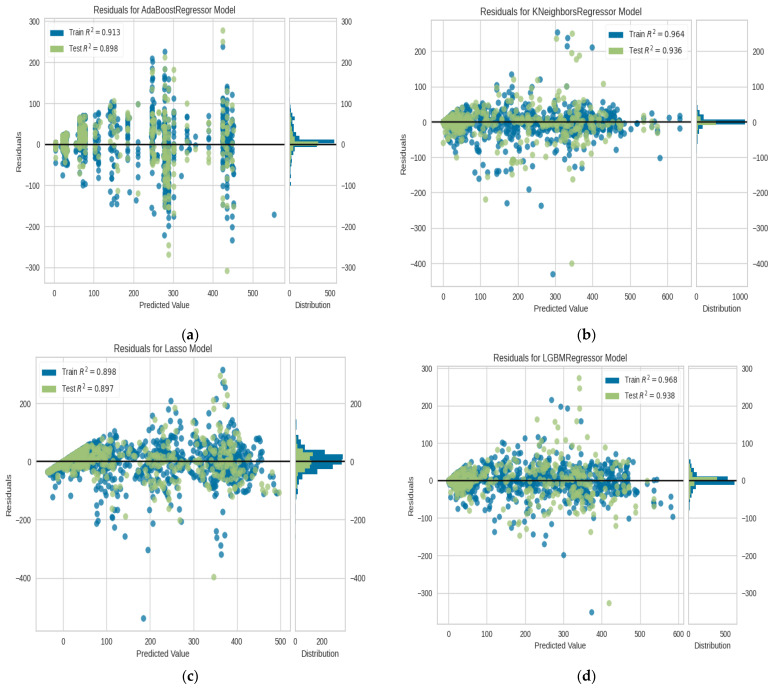
(**a**) Residuals for the Ada-boost regression model, (**b**) residuals for the K-Nearest Neighbors regressor model, (**c**) residuals for the LASSO model, and (**d**) residuals for the LGBM regressor model.

**Figure 14 ijerph-20-05943-f014:**
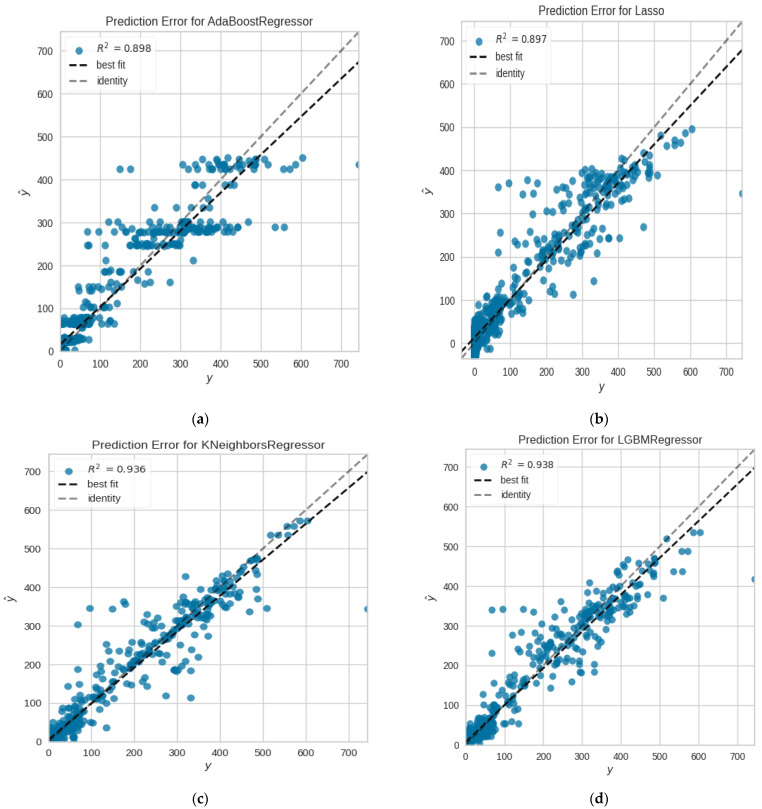
Prediction errors of (**a**) the Ada-Boost regressor, (**b**) the LASSO model, (**c**) the K-Neighbors model, and (**d**) the LGBM model.

**Figure 15 ijerph-20-05943-f015:**
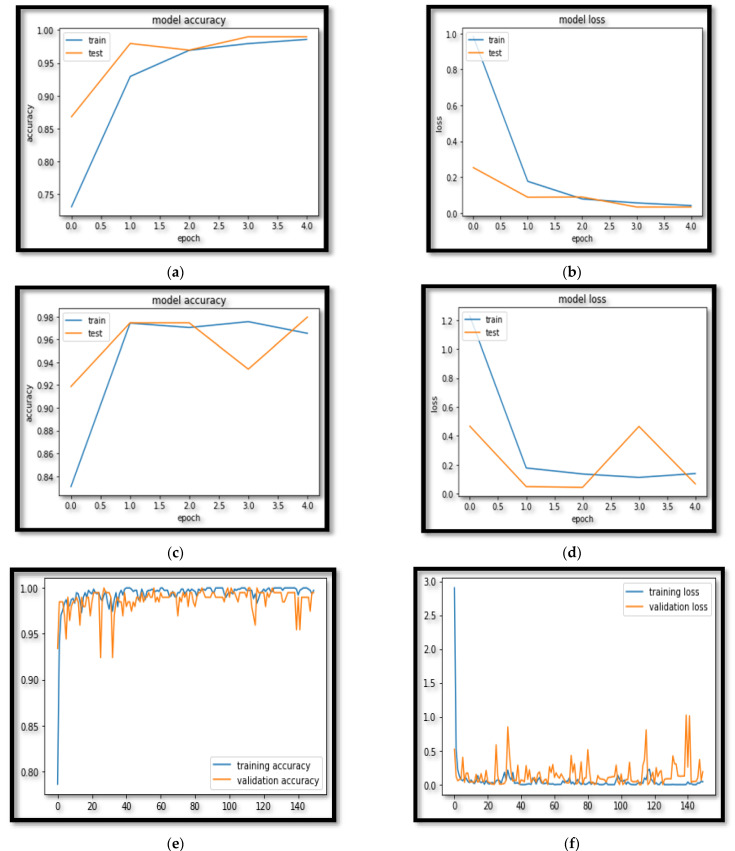
Model accuracy and loss. (**a**) RNN accuracy, (**b**) RNN loss, (**c**) LSTM accuracy, (**d**) LSTM loss, (**e**) AGLSTM accuracy, and (**f**) AGLSTM loss.

**Table 1 ijerph-20-05943-t001:** Review of Literature.

Ref	Methodology Adopted	Data Type	Data Source	Results	Purpose of Research
Kırbas et al. [22]	ARIMA, Nonlinear Autoregression Neural Network (NARNN) and Long-Short Term Memory (LSTM)	Cumulative confirmed cases data of 8 different European countries and the dataset is considered till 3 May 2020	European Center for Disease Prevention and Control	MAPE values of the LSTM model are better than the other models	To model and predict the cumulative confirmed cases and total increase rate of the countries was analyzed and compared. LSTM outperforms other models.
Arora et al. [23]	Deep LSTM/Stacked LSTM, Convolutional LSTM and Bidirectional LSTM	Confirmed cases in India. 14 March 2020 to 14 May 2020	Ministry of Health and Family Welfare	Bi-directional LSTM provides better results than the other models with less error	Daily and weekly predictions of all states are done to explore the increase in positive cases
Chimmula and Zhang [24]	LSTM	confirmed cases in Canada and Italy until 31 March 2020	Johns Hopkins University and Canadian Health Authority	Achieved 92% accuracy	To predict the number of confirmed cases in Canada and Italy and to compare the growth
Shahid et al. [25]	ARIMA, support vector regression (SVR), long short-term memory (LSTM), Bi-LSTM	22 January 2020 to 27 June 2020. 158 samples of the number of confirmed cases, deaths and recovered cases	Dataset is taken from the Harvard University	Bi-LSTM outperforms other models with lower R2 score values	To predict the number of confirmed deaths and recovered cases in 10 countries for better planning and management
Tomar and Gupta [26]	LSTM	Cumulative and daily dataset of COVID-19 cases in India	Center for Systems Science and Engineering (CSSE) at Johns Hopkins University	LSTM achieved 90% accuracy in predicting COVID cases	To predict the number of confirmed and recovered cases using a data-driven estimation method
Shastri et al. [27]	LSTM, Stacked LSTM, Bi-directional LSTM and Convolutional LSTM	India and USA-Confirmed cases data from 7 February to 7 July 2020 Death cases data from 12 March to 7 July 2020.	Datasets of India and USA are taken from the Ministry of Health and Family Welfare, Government of India and Centers for Disease Control and Prevention, U.S Department of Health and Human Services	ConvLSTM outperforms stacked and bi-directional LSTM in confirmed cases and deaths	To predict the number of COVID-19 confirmed and death cases 1 month ahead and to compare the accuracy of deep learning models
Papastefanopoulos et al. [28]	Six different forecasting methods are presented. ARIMA, the Holt-Winters additive model (HWAAS), TBAT, Facebook’s Prophet, Deep AR	January 2020 to April 2020 and the population of countries	Novel Corona Virus 2019 Dataset and population-by-country dataset from kaggle.com	ARIMA and TBAT outperformed other models in forecasting the pandemic	To predict the number of future COVID-19 confirmed death and recovered cases by considering the country’s population
Devaraj, J. et al. [29]	ARIMA, LSTM, Stacked LSTM	22 January 2020 to 8 May 2020. Simulated dataset for seven cities for the months of May, June, July and August 2020. All countries’ data from January 2020 to September 2020	Datasets were collected from John Hopkins University, World Weather Page and Wikipedia page	SLSTM outperformed other models. In statistical analysis, ARIMA outperformed the LSTM model. Overall, the SLSTM model is better than other models.	Global, country-specific, and city-specific cumulative COVID case prediction is done. Feature correlation is done, and the best model prediction is identified through statistical hypothesis testing. Multivariate analysis and prediction of Indian COVID cases are done.
Yahia, N. B. at al. [30]	LSTM, DNN, CNN, Stacked DNN, Stacked LSTM and Stacked CNN	22 January 2020 until 9 November 2020	Datasets were collected from John Hopkins University	Stacked DNN outperformed other models	For the two case studies, China and Tunisia, the stacked-DNN whose inputs are predicted values of LSTM, DNN, and CNN perform better than the stacked LSTM and the stacked CNN
Ayris, D. et al. [31]	The Deep Sequential prediction model (DSPM) and non-parametric regression model (NRM)	22 January to 6 June 2020	Datasets were collected from John Hopkins University	The proposed NRM performed better than the proposed DSPM; however, the difference in performance is not large	
Alassafi, M. O. [32]	RNN and LSTM	Up to 3 December 2020	European Centre for Disease Prevention and Control	The LSTM models	RNN and LSTM
Hawas [33]	Recurrent Neural Network (RNN)	Daily confirmed cases in Brazil 54 to 84 days 7 April to 29 June 2020	Center for Systems Science and Engineering (CSSE) at Johns Hopkins University	Achieved 60.17% accuracy	To predict, 1 month ahead, the confirmed cases and take preventive measures

**Table 2 ijerph-20-05943-t002:** Description of three target variables.

	Cured	Deaths	Cases
Count	1.811000 × 10^4^	18,110.000000	1.811000 × 10^4^
Mean	2.786375 × 10^5^	4052.402264	3.010314 × 10^5^
Std	6.148909 × 10^5^	10,919.076411	6.561489 × 10^5^
Min	0.000000	0.000000	0.000000
25%	3.360250 × 10^3^	32.000000	4.376750 × 10^3^
50%	3.336400 × 10^4^	588.000000	3.977350× 10^4^
75%	2.788698 × 10^5^	3643.750000	3.001498 × 10^5^
Max	6.159676 × 10^6^	134,201.000000	6.363442 × 10^6^

**Table 3 ijerph-20-05943-t003:** Parameters tuned for deep learning models.

S. No.	Hyper-Parameters	Search Space	Type
1	Hidden Layers	[2,10]	Continuous
2	Neurons	[1,100]	Continuous
3	Activation Function	[Tanh and ReLU]	Discrete with step = 1
4	Loss Function	MSE, MAE	Discrete with step = 1
5	Optimizer	Adam, RMS prop	Discrete with step = 1
6	Batch Size	[32,64]	Discrete with step = 1
7	Epochs	[5,200]	Continuous

**Table 4 ijerph-20-05943-t004:** Results of the various metrics used for determining the efficacy of the various prediction models.

Methods	MAE	MSE	RMSE	R2	MAPE	TT (s)
Ada-Boost	32.2941	2543.9772	50.1802	0.8889	11,588.1993	20.23
KNN	22.5399	2205.7787	46.6148	0.9035	14.6692	9.11
Light GBM	22.3603	1987.0647	44.2956	0.9135	106.8839	15.26
LASSO	30.9013	2367.2686	48.2781	0.8969	8707.4533	12.17

**Table 5 ijerph-20-05943-t005:** Results of the various metrics used for determining the efficacy after the hyper-parameter tuning of the various prediction models.

Methods	MAE	MSE	RMSE	R^2^	MAPE	TT (s)
Ada-Boost-_tuned	22.9311	1983.8546	33.4521	0.8999	10,288.1924	17.29
KNN_tuned	19.5649	2014.4652	27.5847	0.9321	145.6382	11.77
LightGBM_tuned	15.5103	1658.2546	37.8796	0.9354	109.8219	14. 70
LASSO_tuned	18.9883	1774.8547	29.7772	0.9215	8103.4151	12.09

**Table 6 ijerph-20-05943-t006:** Results of the different metrics used for determining the efficacy of the various prediction models.

Model	Accuracy	F-Measure	Sensitivity	Specificity	AUC	TT (s)
AGLSTM	99.81 ± 0.21	98 ± 1.04	99 ± 0.85	99 ± 0.72	98 ± 1.02	6.09
LSTM	97.97 ± 1.02	96.87 ± 1.82	97.47 ± 1.05	97.97 ± 1.02	96.97 ± 1.02	11.87
RNN	96.95 ± 1.75	96.95 ± 1.55	95.95 ± 1.75	96.95 ± 1.25	95.95 ± 1.84	18.11

## Data Availability

The data that support the findings of this study are available at indiastathealth.com and covidindia.org.

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
