# Peer review of "Analysis and Prediction of COVID-19 Multivariate Data Using Deep Ensemble Learning Methods"

_ijerph, 2023, doi:10.3390/ijerph20115943_

Round 1
Reviewer 1 Report
Dear Authors
Your text “Analysis and Prediction of Covid19 Multivariate Data using Deep Ensemble Learning Methods” can be definitely acceptable for IJERPH. But please pay attention to my remarks on your article that may help you improve some moments.
After the corrections, the text may be published.
Best regards, the Reviewer
REMARKS:
1. The Abstract should be written as a concise piece of logically completed text without numbering, nor words “Background, Methods, etc. Please add main conclusions to the Abstract? What did you achieve in your research?
2. Line 28: “Some local governments…” Which exactly? Please be more specific.
3. Lines 39-40: “Every infectious disease epidemic has a pattern that can be figured out by looking at how the disease is spreading” had better be removed. It is commonplace.
4. Chapter 1.2. It is VERY difficult to argue about mental disorders in terms of percentages, what is done in the works you cite. There is no clear definition of dementia and other mental impairments in psychiatrics. I advise you to add a proviso about that.
5. Chapter 1.2. Since “New cases of COVID-19…” and by the end of the chapter. I recommend complete removing this paragraph. It has nothing to do with your research. Besides, please do not lay on colours too thickly.
6. Chapter 1.3. Is considering the origin of SARS-CoV-2 relevant to your study? If so, please prove it in the chapter. If no, please remove.
7. Chapter 3. What new information can Spearman and Kendall correlation coefficients bring into your investigation besides the more traditional Pearson’s corr. coeff.? Please explain it in the chapter.
8. In almost all figures please increase the size of captions. With so little size, they are illegible.
9. Please rename Chapter 6. “Results” cannot follow “Discussion” that is Chapter 5!
10. Please provide major conclusions form your work stated very briefly.
11. Please add a separate (sub)chapter “Limitations”.
12. Lines 491-492. “…on the Indian states…” I did not see any comparison among Indian states! One may thing that he can find a comparison between UP and Gujarat from your words. You write about India in total. Am I wrong? Please re-formulate your sentence.
Good luck in improving!
Reviewer 2 Report
The authors propose an adaptive gradient LSTM model to predict the Covid19 disease and provide some comparison results with other machine learning techniques. Two major concerns exist:
1. The compared methods are not the newest and the review is not sure the proposed algorithm can achieve the best performance than SOTA as they claimed as the advantages of their design are not fully explored.
2. Presentation needs extensive improvements. The listed figures and tables are not easy to read.
Minor editing of English language are required but major improvements should be provided in terms of presentation.
Reviewer 3 Report
In this article, the authors focused on the prediction of Covid-19 multivariate data using ML and DL models. The research recommends an adaptive gradient LSTM model (AGLSTM) using multivariate time series data. The RNN, LSTM, Ada-Boost, Light Gradient Boosting and KNN models are also evaluated. The experiment results showed that the output of AGLSTM performs better than the other models in terms of accuracy and R2 and requires less time for training and prediction.
The paper is well organized and has the proper structure and flow. However, it lacks depth and novelty. The technical contribution is limited.
The submission needs critical technical interventions as it has weaknesses in its current form.
1)In the abstract, there should also be numerical results concerning the AGLSTM model.
2)Also, in the abstract (and in many places in the article), you mention the LASSO technique as a model. LASSO is a regression analysis method.
3)Κeywords after the abstract should also include Covid-19.
4)In the introduction, lines 77-87 need documentation.
5) Reference [9] is not mentioned anywhere in the article.
6) At the end of the introduction, you should note the structure of the article.
7)The acronyms of RNN, LSTM, LASSO and KNN should be defined from the introduction.
8)What are the links to the datasets listed in sub-section 3.1?
9)The metrics of sub-section 3.3 need proper documentation.
10)Please add a table mentioning the parameters tuned for different models used in this study.
11)What is the methodology followed to evaluate the models?
12)The authors miss the experiment setup.
13)In line 451, you omitted the zero (0) from the results.
14)Provide a comparative analysis with previous studies in the discussion section based on the same features, datasets, or algorithms. How your approach is better in terms of accuracy?
15)Please note the limitations and the potential issues of this work, emphasizing the application nature of the proposed method in practice. Also, the overall merit of the proposed approach should be highlighted.
16)The Conclusions section is too weak and needs a better connection with the research results from the proposed model of this study.
17)In this submission, authors should clarify how they interpret Deep Ensemble Learning techniques. Please take into account that there are Ensemble ML models, such as the Stacking model of work
https://www.mdpi.com/1424-8220/23/1/40 and DL models, such as LSTM
and RNN of work https://www.sciencedirect.com/science/article/pii/S096007792030415X
The aforementioned papers should be considered by the authors.
The authors should sift the entire article to remove all issues in terms of English writing. This review requires moderate proofreading.
Round 2
Reviewer 1 Report
My remarks were taken into consideration and the text was clearly improved. The paper may be published. No more comments.
Reviewer 2 Report
The authors are suggested to provide a point-to-point response in the response letter and the current presentation is not easy to observe the main changes.
Minor editing of English language required
Reviewer 3 Report
I have no additional remarks on the revised version.
The authors have addressed my concerns.
Minor editing of the English language is required.